# LipVoicer: Generating Speech from Silent Videos Guided by Lip Reading

**Yochai Yemini, Aviv Shamsian, Lior Bracha, Sharon Gannot, Ethan Fetaya**
Faculty of Electrical Engineering
Bar-Ilan University
Israel
`{yochai.yemini,aviv.shamsian,brachal,sharon.gannot,ethan.fetaya}@biu.ac.il`

## Abstract

Lip-to-speech involves generating a natural-sounding speech synchronized with a soundless video of a person talking. Despite recent advances, current methods still cannot produce high-quality speech with high levels of intelligibility for challenging and realistic datasets such as LRS3. In this work, we present *LipVoicer*, a novel method that generates high-quality speech, even for in-the-wild and rich datasets, by incorporating the text modality. Given a silent video, we first predict the spoken text using a pre-trained lip-reading network. We then condition a diffusion model on the video and use the extracted text through a classifier-guidance mechanism where a pre-trained automatic speech recognition (ASR) serves as the classifier. LipVoicer outperforms multiple lip-to-speech baselines on LRS2 and LRS3, which are in-the-wild datasets with hundreds of unique speakers in their test set and an unrestricted vocabulary. Moreover, our experiments show that the inclusion of the text modality plays a major role in the intelligibility of the produced speech, readily perceptible while listening, and is empirically reflected in the substantial reduction of the word error rate (WER) metric. We demonstrate the effectiveness of LipVoicer through human evaluation, which shows that it produces more natural and synchronized speech signals compared to competing methods. Finally, we created a demo showcasing LipVoicer's superiority in producing natural, synchronized, and intelligible speech, providing additional evidence of its effectiveness. Project page and code: https://github.com/yochaiye/LipVoicer

## 1 Introduction

In the lip-to-speech task, we are given a soundless video of a person talking and are required to accurately and precisely generate the missing speech. Such a task may occur, e.g., when the speech signal is completely obfuscated due to background noises. This task poses a significant challenge as it requires the generated speech to satisfy multiple criteria. This includes intelligibility, synchronization with lip motion, naturalness, and alignment with the speaker's characteristics such as age, gender, accent, and more. Another major hurdle for lip-to-speech techniques is the ambiguities inherent in lip motion, as several phonemes can be attributed to the same lip movement sequence. Resolving these ambiguities requires the analysis of lip motion in a broader context within the video.

Generating speech from a silent video has seen significant progress in recent years, partly due to advancements made in deep generative models. Specifically in applications such as text-to-speech and mel-spectogram-to-audio (neural vocoder) (Kong et al., 2021; Kim et al., 2022). Despite these advancements, many lip-to-speech methods produce satisfying results only when applied to datasets with a limited number of speakers, and constrained vocabularies, like GRID (Cooke et al., 2006) and TCD-TIMIT (Harte & Gillen, 2015). Therefore, speech generation for silent videos in-the-wild still lags behind. We found that these methods struggle to reliably generate natural speech with a high degree of intelligibility on more challenging datasets like LRS2 (Afouras et al., 2018a) and LRS3 (Afouras et al., 2018b).

In this paper, we propose *LipVoicer*, a novel approach for producing high-quality speech for silent videos. The first and crucial part of LipVoicer is leveraging a lip-reading model at inference time,

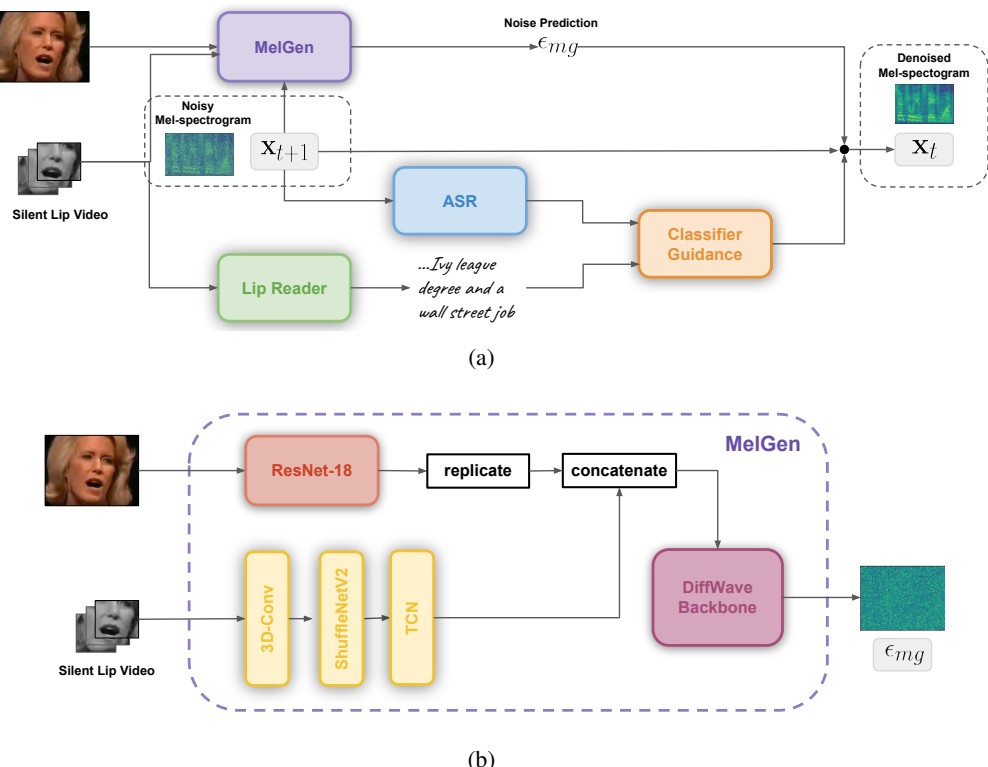

Figure 1: An illustration of LipVoicer, a dual-stage framework for lip-to-speech. (a) To generate the speech from a given silent video at inference time, a pre-trained lip-reader provides additional guidance by predicting the text from the video. An ASR steers MelGen, which generates the mel-spectrogram, in the direction of the extracted text using classifier guidance, such that the generated mel-spectrogram reflects the spoken text. (b) MelGen, our diffusion denoising model that generates mel-spectrograms conditioned on a face image and a mouth region video extracted from the full-face video using classifier-free guidance. It is trained without the text guidance.

for extracting the transcription of the speech we wish to generate. Next, we train a diffusion model, conditioned only on the video, to generate mel-spectrograms. The generation process at inference time is guided by both the video and the predicted transcription. Consequently, our model successfully intertwines the information conveyed by textual modality with the dynamics and characteristics of the speaker, captured by the diffusion model. Incorporating the inferred text has an additional benefit, as it allows LipVoicer to alleviate the lip motion ambiguity to a great extent. Finally, we use the DiffWave (Kong et al., 2021) neural vocoder to generate the raw audio. A diagram of with all of the components in our approach is depicted in Fig. 1 (except the vocoder). Some previous methods often use text to guide the generation process at train time. We, however, utilize it at inference time. The text, transcribed using a lip-reader, allows us to utilize guidance (Dhariwal & Nichol, 2021; Ho & Salimans, 2021) which ensures that the text of the generated audio corresponds to the target text.

We evaluate our LipVoicer model on the challenging LRS2 and LRS3 datasets. These datasets are "in-the-wild" videos, with hundreds of unique speakers and with an open vocabulary. We show that our proposed design leads to the best results on these datasets in both human evaluations as well as WER of an ASR system.

To the best of our knowledge, LipVoicer is the first method to use text inferred by lip-reading to enhance lip-to-speech synthesis. The inclusion of the text modality in inference removes the uncertainty of deciphering which of the possible candidate phonemes correspond to the lip motion. Additionally, it helps the diffusion model to focus on creating naturally synced speech. The speech generated by LipVoicer is intelligible, well synchronized to the video, and sounds natural. Finally, LipVoicer achieves state-of-the-art results for highly challenging in-the-wild datasets.

## 2 BACKGROUND

### 2.1 DENOISING DIFFUSION PROBABILISTIC MODELS (DDPM)

DDPM define a forward process that gradually turns the input into Gaussian noise, then learn the reverse process that tries to recover the input. Specifically, assume a training data point is sampled from the data distribution we wish to model $\mathbf{x}_0 \sim p_{\text{data}}(\mathbf{x})$. The forward process is defined as $q(\mathbf{x}_t|\mathbf{x}_{t-1}) = \mathcal{N}(\mathbf{x}_t|\sqrt{1-\beta_t}\mathbf{x}_{t-1}, \beta_t \mathbf{I})$ where $\{\beta_1, ..., \beta_t, ..., \beta_T\}$ is a pre-defined noise schedule. We can deduce from Gaussian properties that $q(\mathbf{x}_t|\mathbf{x}_0) = \mathcal{N}(\mathbf{x}_t|\sqrt{\bar{\alpha}_t}\mathbf{x}_0, (1-\bar{\alpha}_t)\mathbf{I})$ with $\bar{\alpha}_t = \Pi_{s=1}^t \alpha_s$, and $\alpha_t = 1 - \beta_t$. A sample of $\mathbf{x}_t$ can be obtained by sampling $\epsilon \sim \mathcal{N}(\mathbf{0}, \mathbf{I})$, and using the reparametrization trick, $\mathbf{x}_t = \sqrt{\bar{\alpha}_t}\mathbf{x}_0 + \sqrt{1-\bar{\alpha}_t}\epsilon$. Under mild conditions, the distribution at the final step $q(\mathbf{x}_T)$ is approximately given by a standard Gaussian distribution.

In the reverse process proposed in Ho et al. (2020), $p_\theta(\mathbf{x}_{t-1}|\mathbf{x}_t)$ is then learned by a neural network that tries to approximate $q(\mathbf{x}_{t-1}|\mathbf{x}_t, \mathbf{x}_0)$. In Ho et al. (2020) it was also shown that in order to learn $p_\theta(\mathbf{x}_{t-1}|\mathbf{x}_t)$ it is enough if our model's output $\epsilon_\theta(\mathbf{x}_t, t)$ is trained to recover the added noise $\epsilon$ used to generate $\mathbf{x}_t$ from $\mathbf{x}_0$. The loss function used to train the diffusion model is $\mathbb{E}_{t,\mathbf{x}_0,\epsilon}\left[||\epsilon - \epsilon_\theta(\mathbf{x}_t, t)||^2\right]$. At inference time, given $\mathbf{x}_t$ and the inferred noise, we can sample from $p_\theta(\mathbf{x}_{t-1}|x_t)$ by taking $\mathbf{x}_{t-1} = \frac{1}{\sqrt{\bar{\alpha}_t}}\left(\mathbf{x}_t - \frac{\beta_t}{\sqrt{1-\bar{\alpha}_t}}\epsilon_\theta(\mathbf{x}_t, t)\right) + \beta_t \mathbf{z}$ where $\mathbf{z} \sim \mathcal{N}(0, I)$.

### 2.2 GUIDANCE

One key feature in many diffusion models is the use of guidance for conditional generation. Guidance, both with or without a classifier, enables us to "guide" our iterative inference process to generate outputs that are more faithful to our conditioning information, e.g., in text-to-image, it helps enforce that the generated images match the prompt text.

Assume we wish to sample from $q(\mathbf{x}_t|\mathbf{c})$, $\mathbf{x}_t$ is our sample at the current iteration, $\mathbf{c}$ is some context, and $p(\mathbf{c}|\mathbf{x}_t)$ is a pre-trained classifier. Our goal is to generate $\mathbf{x}_{t-1}$. The idea of classifier guidance (Dhariwal & Nichol, 2021) is to use the classifier to guide the diffusion process to generate outputs that have the right context $\mathbf{c}$. Specifically, if the diffusion model returns $\epsilon_\theta(\mathbf{x}_t, t)$, the classifier guidance alters the noise term that will be used for the update to $\hat{\epsilon} = \epsilon_\theta(\mathbf{x}_t, t) - \omega_1 \sqrt{1 - \bar{\alpha}_t} \nabla_{\mathbf{x}_t} \log p(\mathbf{c}|\mathbf{x}_t)$ where $\omega_1$ is a hyperparameter that controls the level of guidance.

In a later work (Ho & Salimans, 2021), a classifier-free guidance that removes the dependence on an existing classifier is proposed. In classifier-free guidance, we make two noise predictions, one with the conditioning context information, $\epsilon_\theta(\mathbf{x}_t, \mathbf{c}, t)$, and one without it - $\epsilon_\theta(\mathbf{x}_t, t)$. We then use $\hat{\epsilon} = \epsilon_\theta(\mathbf{x}_t, \mathbf{c}, t) + \omega_2(\epsilon_\theta(\mathbf{x}_t, \mathbf{c}, t) - \epsilon_\theta(\mathbf{x}_t, t))$ where the hyperparameter $\omega_2$ controls the guidance strength. This allows us to enhance the update directions that correspond to the context $\mathbf{c}$.

## 3 RELATED WORK

### 3.1 AUDIO GENERATION

Following their success in image generation, diffusion models took a leading role as a neural vocoder and in text-to-speech. Examples include: WaveGrad (Chen et al., 2021), Diff-TTS (Jeong et al., 2021), DiffWave (Kong et al., 2021), Grad-TTS (Popov et al., 2021), PriorGrad (Lee et al., 2021), SaShiMi (Goel et al., 2022), Tango (Ghosal et al., 2023), and Diffsound (Yang et al., 2023).
Other important audio generation models that do not rely on diffusion models include: WaveNet (van den Oord et al., 2016), HiFi-GAN (Su et al., 2020), Tacotron (Wang et al., 2017), VoiceLoop (Taigman et al., 2018), WaveRNN (Kalchbrenner et al., 2018), ClariNet (Ping et al., 2018), GAN-TTS (Bińkowski et al., 2019), Flowtron (Valle et al., 2021), and AudioLM (Borsos et al., 2022).

We note that despite the recent progress, unconditional speech generation quality is still unsatisfactory. Thus, some conditioning, e.g., text or mel-spectrogram, is required for high-quality speech generation. We also note that many audio generation models, and particularly lip-to-speech techniques, adopt a sequential approach. Namely, first generating the mel-spectrogram and then using it to generate the raw audio using a pre-trained vocoder.

## 3.2 VISUAL SPEECH RECOGNITION

The field of lip-to-text, also referred to as lip-reading, has seen significant progress in recent years, and multiple techniques are now available (Ma et al., 2022; Shi et al., 2022; Ma et al., 2023). Impressively, they can cope with adverse visual conditions, such as videos where the mouth of the speaker is only partly frontal, challenging lighting conditions and various languages. Use cases of visual speech recognition include resolving multi-talker simultaneous speech and transcribing archival silent films. In our context, using a powerful lip-reading system can help guide the lip-to-speech process.

## 3.3 LIP-TO-SPEECH SYNTHESIS

The task in this research area is to reconstruct the underlying speech signal from a silent talking-face video. A common approach to lip-to-speech includes a video encoder followed by a speech decoder that generates a mel-spectrogram, that is fed to a neural vocoder to generate the final time-domain audio. These techniques have garnered significant attention due to their potential applications in cases where the speech is missing or corrupted. This may occur, for example, due to strong background noise, or when the speech of a person located in the background of the video recording should be attended.

Preliminary studies such as Lip2Wav (Prajwal et al., 2020), End-to-end GAN (Mira et al., 2022), VCA-GAN (Kim et al., 2021) focused on datasetes with limited vocabularies and a number of speakers. Subsequent works such as SVTS (de Mira et al., 2022) addressed more realistic in-the-wild datasets. Recent techniques tremendously improve performance by using AV-HuBERT (Shi et al., 2022) to represent speech by a set of discrete tokens, which are predicted from the video. In ReVISE (Hsu et al., 2022), the authors use AV-HuBERT architecture to generate the audio from the tokens using HiFi-GAN. Concurrently to our work, speech units are also used in Choi et al. (2023b). Also concurrent to LipVoicer, a diffusion-based method is proposed in Choi et al. (2023a).

In Lip2Speech (Kim et al., 2023), the authors use the ground truth text and a pre-trained ASR as an additional loss during training to try to enforce that the generated speech will have the correct text. We, however, use predicted text at inference time.

## 4 LIPVOICER

This section details our proposed LipVoicer scheme for lip-to-speech generation. Given a silent talking-face video $\mathcal{V}$, LipVoicer generates a mel-spectrogram that corresponds to a high likelihood underlying speech signal. The proposed method comprises three main components:

1. A mel-spectrogram generator (MelGen) that learns to create a mel-spectrogram image from $\mathcal{V}$.
2. A pre-trained lip-reading network that infers the most likely text from the silent video.
3. An ASR system that anchors the mel-spectrogram recovered by MelGen to the text predicted by the lip-reader.

At first, we train MelGen, a conditional denoising diffusion probabilistic models (DDPM) model trained to generate a mel-spectrogram waveform $\mathbf{x}$ conditioned on the video $\mathcal{V}$ without the text modality. We use classifier-free guidance to train MelGen (see Section 2.2). Similar to diffusion-based frameworks in text-to-speech, e.g. Jeong et al. (2021), we use a DiffWave (Kong et al., 2021) residual backbone for MelGen. When considering the representation for $\mathcal{V}$, we wish for our representation to encapsulate all the needed information to generate the mel-spectrogram, i.e. the content (spoken words) and dynamics (accent, intonation) of the underlying speech, the timing of each part of speech, as well as the identity of the speaker, e.g. gender, age, etc. However, we wish to remove all irrelevant information to help train and remove unnecessary computational costs. To this end, $\mathcal{V}$ is preprocessed by creating a cropped mouth region video $\mathcal{V}_L$ and randomly choosing a single full-face image $\mathcal{I}_F$. Notably, $\mathcal{V}$ corresponds to the content and dynamics, and $\mathcal{I}_F$ relates to the speaker characteristics. The mouth cropping was implemented according to the procedure in Ma et al. (2022).

To extract features from $\mathcal{V}_L$ and $\mathcal{I}_F$, we use an architecture similar to the one described in VisualVoice (Gao & Grauman, 2021), an audio-visual speech separation model. For $\mathcal{I}_F$, the face embedding

$\mathbf{f} \in \mathbb{R}^{D_f}$ is computed using ResNet-18 (He et al., 2016) with the last two layers discarded. The lip video $\mathcal{V}_L$ is encoded using a lip-reading architecture (Ma et al., 2021). It is composed of a 3D convolutional layer followed by ShuffleNet v2 (Ma et al., 2018) and then a temporal convolutional network (TCN), resulting in the lip video embedding $\mathbf{m} \in \mathbb{R}^{N \times D_m}$, where $N$ and $D_m$ signify the number of frames and channels, respectively. In order to merge the face and lip video embeddings, $\mathbf{f}$ is replicated $N$ times and concatenated to $\mathbf{m}$, yielding the video embedding $\mathbf{v} \in \mathbb{R}^{N \times D}$, where $D = D_f + D_m$. Next, a DDPM is trained to generate the mel-spectrogram with and without the conditioning on the video embedding $\mathbf{v}$ following the classifier-free mechanism (Ho & Salimans, 2021).

In order to make MelGen perform well in scenarios characterized by an unconstrained vocabulary, at inference time we use the text modality as an additional source of guidance. In general, syllables uttered in a silent talking-face video can be ambiguous, and may consequently lead to an incoherent reconstructed speech. It can therefore be beneficial to harness recent advances in lip-reading and ground the generated mel-spectrogram to the text predicted by a pretrained lip-reading network. The question is how to best include this textual information in our framework. One could simply add it as a global conditioning, similar to $\mathcal{I}_F$; however, this ignores the temporal information in the text. One could also try to align the text and the video, which is a complicated process that would introduce additional errors.

To circumvent the challenge of aligning text with video content, we employ text guidance by harnessing the classifier guidance approach (Dhariwal & Nichol, 2021), similarly to Kim et al. (2022). Using a powerful ASR model, we can compute $\nabla_{\mathbf{x}} \log p(t_{LR}|\mathbf{x})$ needed for guidance, where $t_{LR}$ is the text predicted by a lip-reader. The inferred noise $\hat{\epsilon}$ used in the inference update step of the diffusion model is thus modified by both classifier guidance and classifier-free guidance:

$$\hat{\epsilon} = \epsilon_{mg}(\mathbf{x}_t, \mathcal{V}_L, \mathcal{I}_F, \omega_1) - \omega_2\sqrt{1 - \bar{\alpha}_t}\nabla_{\mathbf{x}_t} \log p(t_{LR}|\mathbf{x}_t), \qquad (1)$$

where $\mathbf{x}_t$ is the mel-spectrogram at time step $t$ of the diffusion inference process, and

$$\epsilon_{mg}(\mathbf{x}_t, \mathcal{V}_L, \mathcal{I}_F, \omega_1) = (1 + \omega_1)\epsilon_\theta(\mathbf{x}_t, \mathcal{V}_L, \mathcal{I}_F) - \omega_1\epsilon_\theta(\mathbf{x}_t) \qquad (2)$$

is the estimated diffusion noise at the output of MelGen, and $\omega_1, \omega_2$ are hyperparameters. Note that we use an ASR rather than audio-video ASR, namely $\mathbf{x}_t$ is used as an input to the speech recognizer while the video signal is discarded in order to encourage the model to focus on audio generation.

In our experiments, we noticed that during the generation process, $\epsilon_{mg}$ was much larger in magnitude compared to $\nabla_{\mathbf{x}_t} \log p(t_{LR}|\mathbf{x}_t)$ which led to difficulties in correctly setting $\omega_2$ and to unsatisfactory mel-spectrogram estimates. To remedy this, we followed Kim et al. (2022) by introducing a gradient normalization factor, i.e. Eq. 1 becomes

$$\hat{\epsilon} = \epsilon_{mg} - \omega_2\gamma_t\sqrt{1 - \bar{\alpha}_t}\nabla_{\mathbf{x}_t} \log p(t_{LR}|\mathbf{x}_t) \qquad (3)$$

where

$$\gamma_t = \frac{||\epsilon_{mg}||}{\sqrt{1 - \bar{\alpha}_t} \cdot ||\nabla_{\mathbf{x}_t} \log p(t_{LR}|\mathbf{x}_t)||}. \qquad (4)$$

and $|| \cdot ||$ is the Frobenius norm. The inference process is summarized in Algorithm 1 in the Appendix.

Classifier guidance allows us to train MelGen that is solely conditioned on $\mathcal{V}$, and use a pre-trained ASR to make the generated speech match $t_{LR}$. As a result, the ASR is responsible for the precise words in the estimated speech, and MelGen provides the voice characteristics, synchronization between $\mathcal{V}$ and $\mathbf{x}$, and the continuity of the speech. One additional advantage of this approach is the modularity and ease of substituting both the lip-to-text and the ASR modules. If one wishes to substitute these models with improved versions in the future, the process can be accomplished effortlessly without requiring any re-training. Finally, a DiffWave vocoder (Kong et al., 2021) is used to transform the reconstructed mel-spectrogram to a time-domain speech signal. Note that the vocoder does not appear in Fig. 1.

## 5 EXPERIMENTS

In this section, we compare LipVoicer to various lip-to-speech approaches. We use multiple datasets and learning setups to evaluate LipVoicer. To encourage future research and reproducibility, our source

code will be made publicly available. Additional ablation studies, experimental results on GRID, and complete details are provided in the Appendix. We also created the following website `https://lipvoicer.github.io` containing sample videos generated by all compared approaches. We believe this is the best way to fully understand the performance gain of LipVoicer, and highly encourage the readers to visit.

**Datasets**    LipVoicer is compared against the baselines on the highly challenging datasets LRS2 (Afouras et al., 2018a) and LRS3 (Afouras et al., 2018b). LRS2 contains roughly 142,000 videos of British English in its train and pre-train splits, which amounts to 220 hours of speech by various speakers. In the test set, there are 1,243 videos. The train and pre-train sets of LRS3 comprise 9,000 different speakers, contributing 151,000 videos which are stretched across 430 hours of speech videos. There are 1,452 videos in the test split. The language spoken in LRS3 videos is English but with different accents, including non-native ones. We specifically select these in-the-wild datasets, LRS2 and LRS3, for their diverse range of real-world scenarios with variations in lighting conditions, speaker characteristics, speaking styles, and speaker-camera alignment. We train LipVoicer using the pretrain+train splits of LRS2 and LRS3 on each dataset separately, and evaluation is carried out on the full unseen test data splits. Note that both datasets also include transcriptions, but we do not use them for either training or inference.

**Implementation Details**    For predicting the text from the silent video at inference time, we use Ma et al. (2023) as our lip-reader for LRS2 and LRS3. It achieves a WER rate of 14.6% and 19.1% with respect to the ground truth text, respectively. The ASR deployed for applying classifier guidance was Burchi & Timofte (2023). We modified its architecture by adding the diffusion time step embedding to the conformer blocks and then fine-tuned on LRS2 and LRS3. Finally, for the sake of fairness, we employ a different ASR (Ma et al., 2023) to evaluate the WER of the speech generated by our method and the baselines. We notice some variability in the choice of the ASRs used for benchmarking among the baselines, which make it difficult to put the different methods on a common ground. We believe that Ma et al. (2023) can serve as a good candidate to be used in future studies, since it achieves state of the art results on LRS2 (1.5%) and LRS3 (1%) and a pre-trained model is publicly available. The rest of the implementation details for reproducing our experiments can be found in the Appendix.

**Baselines**    We compare LipVoicer with recent lip-to-speech synthesis baselines. The baseline methods include (1) SVTS (de Mira et al., 2022), a transformer-based video to mel-spectrogram generator with a pre-trained neural vocoder. (2) VCA-GAN (Kim et al., 2021), a GAN model with a visual context attention module that encodes global representations from local visual features. (3) Lip2Speech (Kim et al., 2023), a multi-task learning framework that incorporates ground truth text to form an additional loss to enforce the correspondence between the text predicted from the generated speech and the target text at train time. We trained VCA-GAN and Lip2Speech on LRS2 and LRS3, and used the LRS3 test files provided by the authors of SVTS to conduct the comparisons.

Unfortunately, we could not include ReVISE (Hsu et al., 2022) in our full comparisons since the test files are unavailable for public access and training the method is too computationally heavy (32 GPUs were reported in ReVISE). Nevertheless, we used the ASR utilized in Hsu et al. (2022) to compare it to LipVoicer. Notably, in Hsu et al. (2022) it was reported that the ASR achieved WER score of 5.6% on the ground-truth test videos of LRS3, but we only managed to achieve 6.4% on the same data. While we are uncertain of the source of the degraded performance, we evaluated LipVoicer using this ASR and achieved WER score of 33.2% on LRS3, whereas ReVISE reported 33.9%. Additionally, we generated with LipVoicer the speech signals for the silent videos on the project page of ReVISE and incorporated the results in our demo website. We note that the speech units computed by AV-HuBERT are primarily learned in such a way that encourages speech continuity, and not any other explicit speech aspects (WER, intelligibility, etc.). We believe that this is the reason why ReVISE still has relatively high WER, although much lower than its previous competitors.

**Metrics**    Several metrics are used to evaluate the quality and intelligibility of our generated speech and compare it to the baseline methods. As the main goal of speech generation is to create a speech signal that sounds natural and intelligible to human listeners, our focus is on human evaluation measured by the mean opinion score (MOS). We also compare WER using a speech recognition model. Following Hsu et al. (2022); Hassid et al. (2021), we measure the synchronization between

|  | Intelligibility | Naturalness | Quality | Synchronization |
|---|---|---|---|---|
| GT | $4.33 \pm 0.04$ | $4.43 \pm 0.04$ | $4.34 \pm 0.04$ | $4.39 \pm 0.04$ |
| LIP2SPEECH (Kim et al., 2023) | $2.07 \pm 0.08$ | $1.98 \pm 0.08$ | $1.93 \pm 0.08$ | $2.66 \pm 0.10$ |
| VCA-GAN (Kim et al., 2021) | $1.77 \pm 0.08$ | $1.85 \pm 0.09$ | $1.77 \pm 0.08$ | $2.34 \pm 0.09$ |
| LIPVOICER (OURS) | $\mathbf{3.53 \pm 0.07}$ | $\mathbf{3.54 \pm 0.08}$ | $\mathbf{3.69 \pm 0.08}$ | $\mathbf{3.82 \pm 0.07}$ |

Table 1: LRS2 Human evaluation (MOS).

|  | Intelligibility | Naturalness | Quality | Synchronization |
|---|---|---|---|---|
| GT | $4.38 \pm 0.03$ | $4.45 \pm 0.03$ | $4.42 \pm 0.03$ | $4.36 \pm 0.03$ |
| LIP2SPEECH (Kim et al., 2023) | $2.21 \pm 0.08$ | $2.20 \pm 0.09$ | $2.01 \pm 0.07$ | $2.69 \pm 0.08$ |
| SVTS (de Mira et al., 2022) | $2.17 \pm 0.08$ | $2.15 \pm 0.09$ | $1.99 \pm 0.07$ | $2.71 \pm 0.09$ |
| VCA-GAN (Kim et al., 2021) | $2.19 \pm 0.08$ | $2.20 \pm 0.09$ | $2.08 \pm 0.08$ | $2.71 \pm 0.08$ |
| LIPVOICER (OURS) | $\mathbf{3.44 \pm 0.07}$ | $\mathbf{3.52 \pm 0.07}$ | $\mathbf{3.42 \pm 0.08}$ | $\mathbf{3.56 \pm 0.07}$ |

Table 2: LRS3 Human evaluation (MOS).

the generated speech and the matching video with SyncNet (Chung & Zisserman, 2016) scores. Specifically, we report the temporal distance between audio and video (LSE-D) and the confidence scores (LSE-C). We use the pre-trained SyncNet model[1] to calculate LSE-C and LSE-D.

To objectively quantify the quality and intelligibility of our scheme, we use DNSMOS (Reddy et al., 2022) and STOI-Net (Zezario et al., 2020). It is worth mentioning that previous studies on lip-to-speech synthesis have presented metrics that we believe are unsuitable for this task, and as a result, we refrained from including them in our main report. Specifically, they use short-time objective intelligibility (STOI) and perceptual evaluation of speech quality (PESQ). Both metrics are intrusive, namely, they are based on a comparison with the clean raw audio signal. While they are valuable for speech enhancement and speaker separation, in speech generation, it is not expected, even for a perfect model, to recreate the original audio. This may stem, for instance, from a variation of the pitch between the original speech and the reconstructed one (Sisman et al., 2020). Our goal is to generate a speech signal that matches the *video*, and not *the original speech*. In Appendix D, we provide an example that demonstrates the effectiveness of non-intrusive over intrusive metrics with regard to the lip-to-speech setup. In this sense, our problem is more related to voice conversion. For completeness, the Appendix also includes the PESQ and STOI scores of LipVoicer.

## 5.1 HUMAN EVALUATION RESULTS

Given 50 random samples from the test splits of LRS2 and LRS3 datasets (each), we used Amazon Mechanical Turk to rate the different approaches and the ground truth. The listeners were asked to rate the videos, on a scale of 1-5, for Intelligibility, Naturalness, Quality, and Synchronization. We note that we observed a non-negligible amount of noise in the scores of the human evaluators. We ameliorated this effect by having a large number (16) of distinct annotators per sample, and by having each annotator score all methods on a specific sample. We also filtered out Turkers who rated the ground-truth video with a score smaller than 4. This ensures that all methods compared are annotated by the exact same reliable set of annotators. We also note that for the SVTS (de Mira et al., 2022) baseline comparison, we used generated videos that were sent to us by the authors. As only LRS3 videos were available, we evaluate SVTS only on LRS3.

Tables 1 and 2 depict the MOS results on the LRS2 and LRS3 datasets. LipVoicer outperforms all three baseline methods: Lip2Speech, SVTS, and VCA-GAN. Not only that, it is also remarkably close to the ground truth scores. The questionnaire, screenshots of the task, and other implementation details are given in Appendix G.

---

[1]`https://github.com/joonson/syncnet_python`

| | WER ↓ | STOI-Net ↑ | DNSMOS ↑ | LSE-C ↑ | LSE-D ↓ |
|---|---|---|---|---|---|
| GT | 1.5% | 0.91 | 3.14 | 6.840 | 7.194 |
| LIP2SPEECH | 51.4% | 0.70 | 2.37 | **6.815** | **7.370** |
| VCA-GAN | 100.7% | 0.51 | 2.26 | 3.369 | 10.703 |
| LIPVOICER (OURS) | **17.8%** | **0.91** | **2.89** | 6.600 | 7.840 |

Table 3: Performance comparison between LipVoicer and the baselines on LRS2.

| | WER ↓ | STOI-Net ↑ | DNSMOS ↑ | LSE-C ↑ | LSE-D ↓ |
|---|---|---|---|---|---|
| GT | 1.0% | 0.93 | 3.30 | 6.880 | 7.638 |
| LIP2SPEECH | 57.4% | 0.67 | 2.36 | 5.231 | 8.832 |
| SVTS | 82.4% | 0.65 | 2.42 | 6.018 | 8.290 |
| VCA-GAN | 90.6% | 0.63 | 2.27 | 5.255 | 8.913 |
| LIPVOICER (OURS) | **21.4%** | **0.92** | **3.11** | **6.239** | **8.266** |

Table 4: Performance comparison between LipVoicer and the baselines on LRS3.

## 5.2 OBJECTIVE EVALUATION RESULTS

We further evaluated our method with the objective metrics. For SVTS, we report WER and synchronization metrics only for LRS3, since the authors did not open-source their code and only released the generated test files for LRS3. From the WER scores, it is clear that our method significantly improves over competing baselines. From the STOI-Net and DNSMOS metrics it is apparent that LipVoicer generates much more intelligible and higher quality speech compared to the competitors. In addition to generating high-quality content, LipVoicer demonstrates commendable synchronization scores, ensuring that the generated speech aligns seamlessly with the accompanying video. Qualitative results may be found in the Appendix.

## 5.3 ABLATION STUDY

The architecture of LipVoicer requires several design choices: the values of $w_1, w_2$, the lip reading network and the ASR used for guidance. We conducted ablation studies examining the influence of the hyperparameters and the modules and evaluated it on LRS3. Tables 5 and 6 show the performance of LipVoicer for different values of $w_1$ and $w_2$, respectively. Several conclusions can be drawn from the results. We observe ($w_1 = -1$) that classifier-free guidance has a decisive role in terms of WER, speech quality, and synchronization. Decreasing the weight of the ASR increases the WER, as expected, while the generated speech still sounds natural and synchronized. It is also clear that ASR guidance is vital, as without it ($w_2 = 0$) the WER plunges from 21.4% to 86.2% on LRS3. Unsurprisingly, an excessive increase in the ASR weight only slightly degrades the WER, but is detrimental to the speech quality and synchronization. Interestingly, setting $w_2 = 1$ or $w_2 = 1.5$ leads to better synchronization compared to $w_2 = 0$ (no ASR guidance). We postulate that this occurs since the predicted text also helps align the generated speech with the video.

With regard to the lip-reader choice, we see in Table 7 that when the ground-truth text is used, the best results are achieved. A less accurate lip-reader leads to worse WER but not speech quality. Synchronization is also impaired with worse lip-reading, probably due to the discrepancy between the lip motion and the predicted text. This ablation study means that by using a more powerful lip-readers, the performance of LipVoicer can be further improved. Ablation studies for the ASR performance, vocoder choice, using the face embedding and the amount of training data can be found in Appendix E.

| $w_1$ | WER ↓ | STOI-Net ↑ | DNSMOS ↑ | LSE-C ↑ | LSE-D ↓ |
|---|---|---|---|---|---|
| -1 | 30.1% | 0.92 | 2.96 | 2.543 | 11.729 |
| 1 | 22.7% | 0.93 | 3.09 | 5.512 | 8.885 |
| **2** | **21.4%** | **0.92** | **3.11** | **6.239** | **8.266** |
| 3 | 22.4% | 0.92 | 3.07 | 6.435 | 7.967 |
| 4 | 22.2% | 0.92 | 3.03 | 4.608 | 9.636 |

Table 5: Ablation study for $w_1$ evaluated on LRS3. The row marked in bold indicates the value of $w_1$ used by LipVoicer in the experimental study for LRS3.

| $w_2$ | WER ↓ | STOI-Net ↑ | DNSMOS ↑ | LSE-C ↑ | LSE-D ↓ |
|---|---|---|---|---|---|
| 0 | 86.2% | 0.93 | 3.16 | 6.318 | 8.310 |
| 1 | 23.5% | 0.93 | 3.12 | 6.556 | 7.870 |
| **1.5** | **21.4%** | **0.92** | **3.11** | **6.239** | **8.266** |
| 3 | 22.3% | 0.91 | 2.87 | 5.189 | 9.156 |
| 4 | 24.4% | 0.89 | 2.71 | 4.608 | 9.636 |

Table 6: Ablation study for $w_2$ evaluated on LRS3. The marked in bold indicates the value of $w_2$ used by LipVoicer in the experimental study for LRS3.

| LR | LR WER | WER ↓ | STOI-Net ↑ | DNSMOS ↑ | LSE-C ↑ | LSE-D ↓ |
|---|---|---|---|---|---|---|
| GT | 0% | 5.4% | 0.92 | 3.10 | 6.257 | 8.220 |
| Ma et al. (2023) | 19.1% | 21.4% | 0.92 | 3.11 | 6.239 | 8.266 |
| Ma et al. (2022) | 32.3% | 38.1% | 0.92 | 3.09 | 6.053 | 8.362 |

Table 7: Ablation study for the choice of the lip reading accuracy, as evaluated on LRS3. LR signifies lip-reader.

## 6    LIMITATIONS AND SOCIAL IMPACTS

LipVoicer is a powerful lip-to-speech method that has the potential to bring about some social impacts. On the positive side, it can help restore missing or corrupt speech in important videos. However, there are also potential drawbacks to consider. The generated speech may introduce the risk of misrepresentation or manipulation. While we use a faithful lip-reading module, "bad-faith actors" can try to inject misleading text. Mitigating this potential risk is an important challenge but beyond the scope of this work.

## 7    CONCLUSION

In this paper, we present LipVoicer, a novel method that shows promising results in generating high-quality speech from silent videos. LipVoicer achieves this by utilizing text inferred from a lip-reading model to guide the generation of natural audio. We train and test LipVoicer on multiple challenging datasets comprised of in-the-wild videos. We empirically show that text guidance is crucial to creating intelligible speech, as measured by the word error rate. Furthermore, we show through human evaluation that LipVoicer faithfully recovers the ground truth speech and surpasses recent baselines in intelligibility, naturalness, quality, and synchronization. The impressive achievements of LipVoicer in lip-to-speech synthesis not only advance the current state-of-the-art but also pave the way for intriguing future research directions in this domain.

ACKNOWLEDGMENTS

This project has received funding from the European Union's Horizon 2020 Research and Innovation Programme, Grant Agreement No. 871245, and was also supported in part by the Israeli Council for Higher Education, Data Science Program ("Audience" project) and Nvidia academic hardware grant.

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

# A  FULL IMPLEMENTATION DETAILS

In this section, we provide full details on the preprocessing, hyperparameters and architectures used for LipVoicer. Our implementation was written in PyTorch, and we used 4 NVIDIA GeForce RTX 2080 Ti for our experiments.

## A.1  VIDEO PREPROCESSING

The output of the preprocessing step of a silent talking-face video is a single face image and a mouth crop video. The latter is used by the lip-reading network, while both components are fed into the MelGen module. We use 25 fps videos. The face image is randomly chosen from the full-face video frames. The image is resized to the resolution of $224 \times 224$. During training, brightness and color augmentations are applied as well. The mouth crop video is created by detecting and tracking 68 facial landmarks using dlib (King, 2009). The faces detected in the frames are then aligned to a mean reference face using a similarity transformation which removes rotation and scale differences. Next, a $96 \times 96$ mouth region is cropped from each frame to form the mouth crop video. The last step is the conversion to a gray-scale video. In the training stage, a random $88 \times 88$ crop is taken from the lip video and then horizontally flipped with probability 0.5. At inference time, we take a center $88 \times 88$ crop without flipping.

## A.2  AUDIO PREPROCESSING

In LipVoicer, we need mel-spectrograms of real speech signals for training MelGen and the vocoder. They were computed by applying the following steps. First, 16KHz speech signals were extracted from the training videos of the benchmark datasets and normalized to have a maximum magnitude of 0.9. Subsequently, the short-time Fourier transform (STFT) for all experiments was calculated with window size and Discrete Time Fourier Transform (DTFT) length of 640 samples. The hop size between two successive windows was 160 samples. The STFT magnitude was used to compute 80 mel frequencies with 20Hz and 8KHz being the lowest and highest frequencies, respectively. The result was clipped to have a minimum value of $1e - 5$. The logarithm function was applied to compress the dynamic range. Finally, we linearly mapped the output of the previous step to roughly be in $[-1, 1]$ by using the global minimum and maximum values across the entire training split.

## A.3  ARCHITECTURE

### A.3.1  MELGEN

**Backbone.**  The backbone architecture for MelGen is identical to DiffWave (Kong et al., 2021) but we change several parameters. Using the notations from Kong et al. (2021), the diffusion process has $T = 400$ steps. We set $\beta_1 = 0.0001$ and $\beta_T = 0.02$, and use a linear noise schedule. The number of input and output channels is fixed to 80, to match the number of mel-spectrogram frequencies. We use 12 residual layers with 512 residual channels, and the dilation factor was set to 1 for all residual layers. We base our backbone architecture on the publicly available implementation[2] of DiffWave, released by the authors of Goel et al. (2022).

**Conditioner.**  As mentioned, the video embedding serves as the conditioner for MelGen. The face image analysis network takes in a $224 \times 224$ image and comprises a ResNet-18 backbone with the last two layers scrapped, followed by a fully-connected layer which outputs a $D_f = 128$ dimensional embedding vector. The lip motion analysis network receives a $88 \times 88$ mouth crop of $N$ consecutive frames. It consists of a 3D-convolutional layer with a $5 \times 7 \times 7$ kernel, followed by a ShuffleNet-V2 (Ma et al., 2018) network and then a TCN. Full details can be found in Ma et al. (2021). The output of the TCN takes the dimension $N \times D_m$ where $D_m = 512$. The face embedding is replicated $N$ times and concatenated to the lip motion embedding, such that the ultimate video embedding **v** has the size of $N \times 640$. Similarly to Kong et al. (2021), we upsample the time axis of the video embedding by applying two layers of 2D convolutions. As each mel-spectrogram frame comprises 160 new samples, the audio sample rate is 16KHz and videos are sampled at 25 fps, we set the upsampling factor of each convolution layer to 2. Under this choice, the overall upsampling factor amounts to 4 which aligns the time resolution of the mel-spectrogram and the video.

---

[2]`https://github.com/albertfgu/diffwave-sashimi`

**Training.** During training, MelGen generates mel-spectrograms covering 1 second conditioned on the matching video with the equivalent number of frames. It translates to 100 mel-spectrogram frames and hence the video comprises $N = 25$ frames. Note that at inference time, since the DiffWave backbone is based on convolutional layers, we take a video of an arbitrary number of frames. In our experiments, we managed to generate high-quality speech signals which were 18 seconds long. MelGen is trained with $L_1$ loss on the diffusion noise prediction, without any additional loss terms. We trained on 1,000,000 mini-batches of 16 videos and used Adam optimizer with learning rate of $2e - 4$ without scheduling. For the classifier-free guidance mechanism, we follow Ho & Salimans (2021) by setting the dropout probability on the conditioning to 0.2. The null tokens for the face image and the lip motion video when the conditioning is dropped are randomly drawn in advance from a normal distribution.

### A.4 ASR CLASSIFIER

As mentioned, Burchi & Timofte (2023) was deployed as the ASR for guiding MelGen through the classifier guidance mechanism. We used the publicly available code and the pre-trained model released by the authors of Burchi & Timofte (2023), which was trained on the LRW, LRS2, and LRS3 datasets. In order to adapt their implementation to be compatible with LipVoicer, we added a diffusion time step embedding to the conformer blocks. The computation of the time step embedding is identical to Kong et al. (2021), i.e. we use the same positional encoding and two fully-connected layers which are shared across all conformer blocks. Each conformer applies an additional layer-specific fully-connected layer, and the result is summed with the input to the conformer block. In addition, since for LipVoicer, the ASR classifier receives mel-spectrograms noised by the diffusion process, we remove the mel-spectrogram and SpecAugment (Park et al., 2019) layers from the original implementation of the ASR. We finetune our modified ASR on the train+pretrain splits of LRS2 and LRS3 using connectionist temporal classification (CTC) loss, where the input is a mel-spectrogram from the training splits noised by the diffusion process and the target is the ground truth text. We used Adam optimizer with learning rate $1e - 4$ and akin to Burchi & Timofte (2023), we used batch size 0f 256 videos via mini-batches of size 64 and 4 accumulated steps.

## B INFERENCE

We summarize the inference process in Algorithm 1

---

**Algorithm 1** Inference with LipVoicer

---

1: Given a silent video $\mathcal{V}$
2: $\mathcal{V}_L \leftarrow$ lip video
3: $\mathcal{I}_F \leftarrow$ randomly chosen face frame from $\mathcal{V}$
4: Predict the text $t_{LR}$ from $\mathcal{V}_L$ with a lip-reader
5: Initialize $\mathbf{x}_T \sim \mathcal{N}(\mathbf{0}, \mathbf{I})$
6: **for** $t = T \dots 1$ **do**
7:     Compute $\epsilon_{mg}$ with classifier-free guidance
8:     $\epsilon_{mg}(\mathbf{x}_t, \mathcal{V}_L, \mathcal{I}_F, \omega_1) = (1 + \omega_1)\epsilon_\theta(\mathbf{x}_t, \mathcal{V}_L, \mathcal{I}_F) - \omega_1\epsilon_\theta(\mathbf{x}_t)$
9:     Compute $\hat{\epsilon}$ using classifier guidance
10:     $\gamma_t = \frac{||\epsilon_{mg}||}{\sqrt{1-\bar{\alpha}_t} \cdot ||\nabla_{\mathbf{x}_t} \log p(t_{LR}|\mathbf{x}_t)||}$
11:     $\hat{\epsilon} = \epsilon_{mg} - \omega_2\gamma_t\sqrt{1 - \bar{\alpha}_t}\nabla_{\mathbf{x}_t} \log p(t_{LR}|\mathbf{x}_t)$
12:     $\mathbf{z}_T \sim \mathcal{N}(\mathbf{0}, \mathbf{I})$ if $t > 1$, else $\mathbf{z} = \mathbf{0}$
13:     $\mathbf{x}_{t-1} = \frac{1}{\sqrt{\bar{\alpha}_t}}\left(\mathbf{x}_t - \frac{\beta_t}{\sqrt{1-\bar{\alpha}_t}}\hat{\epsilon}\right) + \beta_t\mathbf{z}$
14: **end for**
15: **return** $\mathbf{x}_0$

---

## C GENERATING FULL DATASETS

In the course of our experiments, we noticed that different choices of hyperparameters influenced the quality of the reconstructed speech. In particular, the hyperparameters of LipVoicer are the values

of $\omega_1, \omega_2$ and $t_{ASR}$, where the latter denotes the diffusion time step of MelGen in which we start to apply the classifier guidance mechanism. When the ASR guidance was taken into account right from the start ($t_{ASR} = 400$), it led to degraded results. By using a hyperparameter grid search and listening tests, we eventually chose:

- $\omega_1 = 2, \omega_2 = 1.5, t_{ASR} = 270$ for LRS3.
- $\omega_1 = 1.8, \omega_2 = 1.7, t_{ASR} = 230$ for LRS2.
- $\omega_1 = 2.6, \omega_2 = 0.7, t_{ASR} = 300$ for GRID.

## C.1 VOCODER

We adopt DiffWave (Dhariwal & Nichol, 2021) as our vocoder (converting mel-spectrograms to speech signals). We use the conditional variant of DiffWave with $T = 50$ (DiffWave$_{\text{BASE}}$) without changing the parameters of the architecture. We train the vocoder on 1,000,000 mini-batches with batch size 16 with a learning rate of $2e - 4$ without scheduling on audio taken from the training split of LRS3 only.

## D  INTRUSIVE METRICS FOR LIP-TO-SPEECH

Following Sisman et al. (2020), we argue that using intrusive measures like PESQ and STOI in our problem is fundamentally incorrect. To exemplify it, we show that if we take a recording of a certain speaker, they cannot differentiate between the a noisy version of the recording and the same recording with the change of a speaker. To this end, we use the the following experiment: we took a speech signal $x$ and created two versions of it. One is $y = x + n$, where $n$ is a Gaussian noise with SNR=5dB. Second is $z$, the output of a voice conversion system (Park et al., 2023) where the input is $x$. In other words, $z$ is identical to $x$ with respect to the spoken words and their timing, but with a different yet fairly close voice. We computed the intrusive metrics PESQ, STOI, ESTOI and the non-intrusive ones DNSMOS and STOI-Net and received the results compiled in Table 8.

The intrusive metrics consider the voice converted version as equivalent to the highly noisy signal $y$, which is obviously wrong. For completeness, we also bring here the PESQ and STOI scores for LipVoicer and the baselines for LRS2 (Table 9) and LRS3 (Table10).

## E  ADDITIONAL ABLATION STUDIES

### E.1  ASR WER AND FACE EMBEDDING

Evaluating the performance of LipVoicer with respect to the WER of the guiding ASR is presented in Table 11. It is clear that it is necessary for the ASR to be reliable. Table 12 provides objective metrics for the ablation study on using the face embedding. We implemented the ablated model by replacing the face embedding by the null embedding which is part of the classifier-free guidance mechanism. The full model is slightly better, possibly because the model was not trained when only the face image was replaced with the null embedding. The full model replaces the lip region video and the face image with null tokens simultaneously during training. In any case, the main aspect in which discarding of face embedding is manifested is the lack of personalized voice for each video.

|                 | PESQ | STOI | ESTOI | STOI-Net | DNSMOS |
|-----------------|------|------|-------|----------|--------|
| $x$ (clean)     | -    | -    | -     | 0.86     | 3.31   |
| $y$ (noisy)     | 1.38 | 0.74 | 0.52  | 0.74     | 2.45   |
| $z$ (converted) | 1.14 | 0.76 | 0.56  | 0.92     | 3.15   |

Table 8: Intrusive and non-intrusive metrics for the example showcasing why using the former in lip-to-speech should be avoided.

|  | PESQ | STOI | ESTOI | STOI-Net | DNSMOS |
|---|---|---|---|---|---|
| GT | - | - | - | 0.91 | 3.14 |
| LIPVOICER (OURS) | 1.10 | 0.38 | 0.23 | 0.91 | 2.89 |
| LIP2SPEECH | 1.36 | 0.53 | 0.34 | 0.70 | 2.37 |
| VCA-GAN | 1.24 | 0.40 | 0.13 | 0.51 | 2.26 |

Table 9: Intrusive and non-intrusive scores for LRS2.

|  | PESQ | STOI | ESTOI | STOI-Net | DNSMOS |
|---|---|---|---|---|---|
| GT | - | - | - | 0.93 | 3.30 |
| LIPVOICER (OURS) | 1.10 | 0.38 | 0.21 | 0.92 | 3.11 |
| SVTS | 1.26 | 0.53 | 0.31 | 0.65 | 2.42 |
| LIP2SPEECH | 1.31 | 0.50 | 0.27 | 0.67 | 2.36 |
| VCA-GAN | 1.24 | 0.47 | 0.21 | 0.63 | 2.27 |

Table 10: Intrusive and non-intrusive scores for LRS3

We have uploaded to the demo page several examples which compare audio generated by the full and ablated models.

## E.2 VOCABULARY SIZE

We also wish to test the influence of the amount of training data, which entails a larger vocabulary. For LRS2 and LRS3, the vocabulary of the pretrain split is 2-3 times larger than the train split vocabulary. To this end, we conduct a MOS evaluation on LRS3 that compares LipVoicer which was trained on one of the following options: train+pretrain splits or the train split only. In addition, we test LipVoicer coupled with the following lip-reading networks and WER rates: (1) the ground-truth text (2) MA (Ma et al., 2023), achieving $19.1\%$. Comparing to the ground-truth text can better help understand the potential of our proposed method. We only use here the MOS test, since it is important to confirm that the audio quality is good before conducting objective tests which are covered in the main text.

For the ablation study, we randomly select 50 samples from the LRS3 test split and assign three different annotators per sample. Participants are presented with all methods at once when the order of appearance is randomized between samples (see Fig. 4). The average scores are presented in Table 13, and seem to indicate that a larger vocabulary does indeed leads to better performance. When only the train split used for training, using the ground-truth or predicted text results in equivalent performance.

We also examined how the model's influence is affected by increasing the amount of training data even further. We thus trained LipVoicer on LRS3 and videos from VoxCeleb2 (Chung et al., 2018) which contain English speech. The number of training videos amounted to roughly 830,000. It can be seen from Table 14 that the enhanced training data leads to better results.

## E.3 USING A DIFFERENT VOCODER

We compare here the metrics for LRS3 as computed on speech signals generated using HiFi-GAN (Su et al., 2020) and DiffWave as the vocoders. Both vocoders used the exact same mel-spectrograms generated by LipVoicer. Both vocoders used the exact same mel-spectrograms generated by LipVoicer. Also note that the vocoders were trained on natural audio signals on not on mel-spectrograms generated by LipVoicer. We see in Table 15 that both options that use two different vocoders yield state-of-the-are results. This clearly indicates that the main and critical source of improvement is the mel-spectrograms generated by our method.

| ASR WER | WER ↓ | STOI-Net ↑ | DNSMOS ↑ | LSE-C ↑ | LSE-D ↓ |
|---------|-------|------------|----------|---------|---------|
| 3.6%    | 21.4% | 0.92       | 3.11     | 6.239   | 8.266   |
| 6.1%    | 76.3% | 0.92       | 2.98     | 7.118   | 7.171   |

Table 11: Ablation study for the ASR guidance evaluated on LRS3. Inferior ASR WER achieved by early stopping.

| Face Embedding | WER ↓ | STOI-Net ↑ | DNSMOS ↑ | LSE-C ↑ | LSE-D ↓ |
|----------------|-------|------------|----------|---------|---------|
| with           | 21.4% | 0.92       | 3.11     | 6.239   | 8.266   |
| w/o            | 22.9% | 0.92       | 3.05     | 6.091   | 8.407   |

Table 12: Ablation study for using face embedding evaluated on LRS3.

### E.4 QUALITATIVE RESULTS

We present a qualitative comparison between the mel-spectrograms generated by LipVoicer, the baselines, and the ground truth. The results, shown in Fig. 2, demonstrate that even for a relatively long utterance, the mel-spectrogram generated by LipVoicer visibly resembles that of the original video. While all approaches manage to successfully detect the beginning and the end of speech segments, LipVoicer is the only method that generates spectral content that is precisely aligned with the ground truth mel-spectrogram. The baselines, on the other hand, struggle to achieve this level of fidelity. Particularly, they fail to generate pitch information of the speech, which results in an unnatural voice and impairs the naturalness of the speech signal. This comparison provides valuable insights into the capabilities of LipVoicer to generate naturally looking spectrograms that are synchronized with the original video's mel-spectrograms.

### E.5 NATURALNESS-INTELLIGIBILITY TRADE-OFF

Another aspect of LipVoicer that should be taken into consideration is the influence of the ASR classifier on the quality of lip-to-speech performance. Occasionally, the speech signal recovered by MelGen, namely without the classifier guidance provided by the ASR, sounds more natural than the one generated by the full scheme, i.e. LipVoicer. In addition, the ASR guidance may lead to synchronization lapses when the predicted text does not significantly overlap with the ground truth text. However, as emerges from the results in Table 6, the inclusion of the ASR guidance has a crucial impact on the intelligibility of the speech. In particular, it results in an invaluable gain on the WER metric. This trade-off is also depicted in Fig. 3. The mel-spectrogram produced without applying classifier guidance presents smooth transitions. In this context, incorporating the ASR in the inference process degrades the mel-spectrogram, as can be seen by comparing Fig. 3b and Fig. 3c. However, the ASR guidance managed to reconstruct spectral content (right-hand side of the mel-spectrograms) more faithfully with respect to the setup where the ASR guidance is omitted. Consequently, while it may sometimes reduce the speech naturalness, in most cases the generated speech at the output of our method is of high quality, sounds natural, and is intelligible.

## F RESULTS FOR GRID DATASET

In this section, we evaluate LipVoicer on the GRID dataset (Cooke et al., 2006). GRID is an audio-visual dataset comprising 33 speakers, where each speaker contributes 1,000 utterances. We adhere to the data split dictated by the lip-reading network (Ma et al., 2022). Since the code implementation for SVTS (de Mira et al., 2022) is not publicly available and the implementation for Lip2Speech (Kim et al., 2023) required significant modifications, we compare only to VCA-GAN (Kim et al.,

| Source of text / Lip-Reader | Training dataset split | Intelligibility | Naturalness | Quality | Synchronization |
|---|---|---|---|---|---|
| Ground-Truth Text | train | $3.18 \pm 0.28$ | $3.29 \pm 0.28$ | $3.07 \pm 0.25$ | $3.36 \pm 0.26$ |
| Ground-Truth Text | train + pretrain | $4.14 \pm 0.07$ | $4.29 \pm 0.09$ | $4.18 \pm 0.07$ | $4.14 \pm 0.07$ |
| MA (Ma et al., 2023) | train | $3.21 \pm 0.24$ | $3.39 \pm 0.24$ | $3.43 \pm 0.25$ | $\mathbf{3.46 \pm 0.22}$ |
| MA (Ma et al., 2023) | train + pretrain | $\mathbf{3.68 \pm 0.20}$ | $\mathbf{3.64 \pm 0.19}$ | $\mathbf{3.79 \pm 0.21}$ | $3.43 \pm 0.21$ |

Table 13: Ablation study on the amount of training data, as evaluated on LRS3 dataset.

| Training split | WER ↓ | STOI-Net ↑ | DNSMOS ↑ | LSE-C ↑ | LSE-D ↓ |
|---|---|---|---|---|---|
| LRS3 | 21.4% | 0.92 | 3.11 | 6.239 | 8.266 |
| LRS3+VoxCeleb2 | 20.6% | 0.93 | 3.20 | 6.312 | 8.160 |

Table 14: Performance on LRS3 when training on LRS3+VoxCeleb2

2021). We evaluate in terms of MOS using the unseen split[3], where 29 speakers are used for training and the rest (subjects 1,2, 20, 22) are reserved for the test split.

We randomly selected a total of 50 samples from 4 different unseen speakers. Each sample is ranked by 16 unique raters for quality, synchronization, intelligibility, and naturalness, on a scale from 1 to 5. The MOS scores are presented in Table 16. While VCA-GAN managed to achieve good synchronization between the generated audio and the video, LipVoicer outperforms it on all other MOS benchmarks. In particular, we achieve a high level of naturalness and quality and the overall performance is very close to the ground truth speech scores. Audio samples which were *randomly* chosen can be found at `https://lipvoicer.github.io/supp.html`.

## G  PROTOCOL FOR MOS EVALUATION

We used the Amazon Mechanical Turk (AMT) platform to compare the different baseline methods to ours. Participants were asked to rate the videos, on a scale of 1-5, for Intelligibility, Naturalness, Quality, and Synchronization. Fig. 4 shows the task interface, where we simultaneously display to a single annotator the samples generated by (i) LipVoicer (ii) SVTS (iii) VCA-GAN (iv) Lip2Voice, as well as the (v) ground-truth video for calibration. The *order* of the methods is randomized from sample to sample to avoid bias. Each sample is evaluated by 16 distinct annotators. Having each annotator score all methods on a specific sample ensures that all methods compared are annotated by the exact same set of annotators. We take 50 random samples from the test splits of each dataset, LRS3, and LRS2.

The instruction for the task read as follows: *Intelligibility* evaluates how clear words in the synthetic speech sound. In other words, Can you understand the content easily? *Naturalness* evaluates how natural the speech is, compared to the actual human voice (e.g. synthetic speech might sound metallic or distorted). We say the sample is *synced* if there is no time lag between the video and the audio. Finally, *Quality* is the overall quality score for the speech signal given by the rater.

---

[3]`https://github.com/mpc001/Visual_Speech_Recognition_for_Multiple_Languages/tree/master/benchmarks/GRID/labels`

| Vocoder | WER ↓ | STOI-Net ↑ | DNSMOS ↑ | LSE-C ↑ | LSE-D ↓ |
|---|---|---|---|---|---|
| DiffWave | 21.4% | 0.92 | 3.11 | 6.239 | 8.266 |
| HiFi-GAN | 21.9% | 0.93 | 3.19 | 6.308 | 8.166 |

Table 15: Vocoder choice ablation

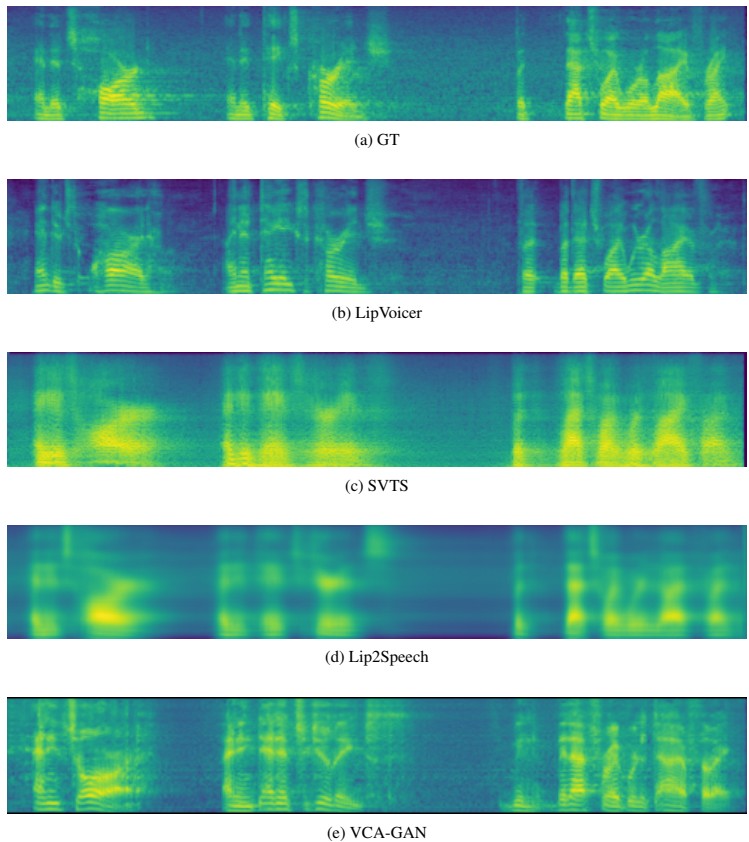

Figure 2: The ground truth mel-spectrogram, and its reconstruction by LipVoicer and the baselines.



Figure 3: The influence of using an ASR for classifier guidance for lip-to-speech with LipVoicer.

| | GRID | | | |
|---|---|---|---|---|
| | Intelligibility | Naturalness | Quality | Synchronization |
| GT | $4.43 \pm 0.06$ | $4.46 \pm 0.06$ | $4.29 \pm 0.06$ | $4.26 \pm 0.05$ |
| VCA-GAN (Kim et al., 2021) | $3.29 \pm 0.15$ | $3.23 \pm 0.17$ | $2.98 \pm 0.14$ | $3.68 \pm 0.11$ |
| LIPVOICER (OURS) | $\mathbf{3.75 \pm 0.11}$ | $\mathbf{3.98 \pm 0.12}$ | $\mathbf{3.60 \pm 0.11}$ | $\mathbf{3.82 \pm 0.11}$ |

Table 16: GRID Human evaluation (MOS).

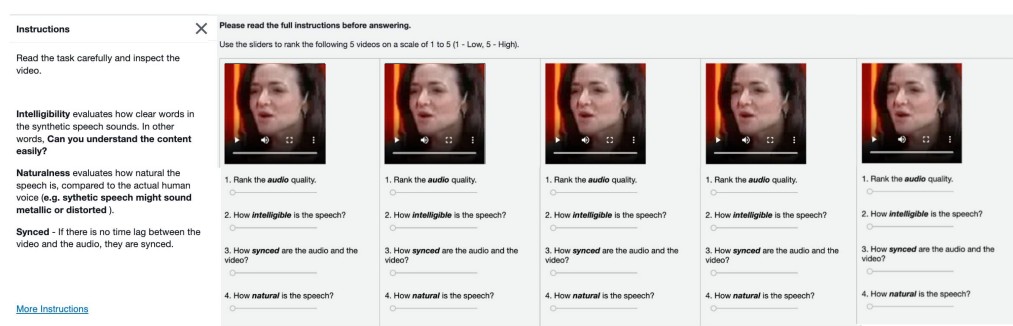

Figure 4: MOS evaluations. Participants rank the samples generated by VCA-GAN (Kim et al., 2021), SVTS (de Mira et al., 2022), LIP2SPEECH (Kim et al., 2023), and LIPVOICER (OURS), and the ground-truth. The order of appearance is randomized.

