# OpenReview forum: "LipVoicer: Generating Speech from Silent Videos Guided by Lip Reading"
_ICLR.cc/2024/Conference — ICLR 2024 poster_

### Official Review · Reviewer_nskp · 2023-10-26

**Soundness:** 3 good
**Presentation:** 3 good
**Contribution:** 3 good
**Rating:** 8
**Confidence:** 4

**Summary:**

This paper describes a new lip-to-speech method that leverages diffusion models via classifier and classifier-free guidance to reproduce accurate speech from silent videos. The base diffusion model is built upon the DiffWave architecture but generates mel-spectrograms instead. It receives a video of the speaker's mouth (encoded into feats via a lipreading backbone) and a still frame representing the speaker's identity (encoded via a simple ResNet) as the condition for generation. This condition is randomly removed during training to perform classifier-free guidance during inference, as proposed in many other diffusion models. After this is trained, the model is further guided via classifier guidance during inference so that the text extracted by a lip reading model from the video matches the text extracted via a speech recognition model from the generated audio. This model achieves SOTA performance on LRS2 and LRS3, and the design decisions are justified by a set of thorough ablations. Demos are also provided, which help contextualize these results empirically.

**Strengths:**

In general, I believe this paper is strong. It clearly sets a new state-of-the-art for lip-to-speech, which is a highly competitive field.

I think the paper is well-written and the motivation for the task and each specific decision in the methodology is concise and meaningful. The methodology is clear and the discussion around the results is welcome. The model figure is adequate in my opinion, and the demos are also very welcome.

The method here is clearly novel - I don't think I've seen a lip-to-speech paper before with a similar architecture and that leverages classifier guidance from text in such an effective way. The choice for each model component makes sense and the training hyperparameters are described in detail to aid reproducibility.

The results are clearly strong and are compared with other works via subjective and objective metrics, which are very convincing. The ablations in tables 5, 6, and 7 are insightful and provide some further information about the importance of the weight of the classifier-free (w1) and classifier (w2) guidance, as well as the lip reading model that is used for the classifier guidance. The avoidance of intrusive measures such as PESQ or STOI is well-justified.

The limitations and social impacts are well addressed, and the conclusions are succinct and valid.

**Weaknesses:**

First and foremost, it is unfortunate that the authors do not compare directly with ReVISE in their tables, although this is fully justified by the lack of code, difficulty in reproducing their results from scratch, and the lack of samples for comparison. Therefore, I don't think it's fair for me or other reviewers to let this affect our judgment of the paper, as this is not the authors' fault. The presented model seems to compare favorably against ReVISE in the demos, which is encouraging.

The use of the DiffWave vocoder is reasonable, but it seems to be outperformed by HiFi-GAN and especially the recent BigVGAN. Would be interesting to see a comparison with these, or at least to justify why DiffWave was chosen, as HiFI-GAN is the typical choice in the majority of papers.

It would also be interesting to scale the model to larger datasets such as LRS3+VoxCeleb2 as was done in SVTS. This would help demonstrate the model's scalability to larger datasets, which is an important aspect of models in this field since audio-visual data is so plentifully available.

I could not find any typos or substantial errors in the writing.

**Questions:**

The authors mention " To encourage future research and reproducibility, our source code will be made publicly available". Will this include training code, inference code, and pre-trained models? These would all be hugely helpful to the community in reproducing the authors' state-of-the-art results.

---

> ### Author Response · Authors · 2023-11-16
>
> 1. *In general, I believe this paper is strong. It clearly sets a new state-of-the-art for lip-to-speech, which is a highly competitive field.*
>
> **Response:**
>
> We thank the reviewer for pointing out that our method currently achieves state-of-the-art results.
>
> 2. *First and foremost, it is unfortunate that the authors do not compare directly with ReVISE in their tables, although this is fully justified by the lack of code, difficulty in reproducing their results from scratch, and the lack of samples for comparison. Therefore, I don't think it's fair for me or other reviewers to let this affect our judgment of the paper, as this is not the authors' fault. The presented model seems to compare favorably against ReVISE in the demos, which is encouraging.*
>
> **Response:**
>
> We thank the reviewer for making this observation. We did our best to make the comparisons to ReVISE that were possible given the circumstances.
>
> 3. *The use of the DiffWave vocoder is reasonable, but it seems to be outperformed by HiFi-GAN and especially the recent BigVGAN. Would be interesting to see a comparison with these, or at least to justify why DiffWave was chosen, as HiFI-GAN is the typical choice in the majority of papers.*
>
> **Response:**
>
> We chose to use DiffWave as our vocoder since we had already used its architecture for MelGen, so applying a DiffWave vocoder was immediate.
> Since Reviewer gU1H has raised the argument that LipVoicer outperforms the baseline thanks to the vocoder it uses, we are now training also a HiFi-GAN vocoder and once training is completed we will use it to synthesise test audio samples and upload the calculated metrics to this page. HiFi-GAN was used in ReVISE, for instance.
>
> 4. *It would also be interesting to scale the model to larger datasets such as LRS3+VoxCeleb2 as was done in SVTS. This would help demonstrate the model's scalability to larger datasets, which is an important aspect of models in this field since audio-visual data is so plentifully available.*
>
> **Response:**
>
> We have been downloading the VoxCeleb2 dataset over the last few days, and have been experiencing a slow download speed of tens to hundreds of KB/s. From our diagnosis, the source of the slow speed is on the server side. Since VoxCeleb2 is a big dataset and the downloaded videos have to undergo pre-processing stages (audio extraction etc), we cannot be assured that we will manage to carry out the experiments with VoxCeleb2 by the end of the discussion period. Instead, in the meantime, we are conducting this experiment with the concatenation of LRS3 and LRS2, including their pretrain splits. We will upload the results when they are ready, and will add the experiment with VoxCeleb2 to the camera ready version of this paper.
>
> 5. *The authors mention " To encourage future research and reproducibility, our source code will be made publicly available". Will this include training code, inference code, and pre-trained models? These would all be hugely helpful to the community in reproducing the authors' state-of-the-art results.*
>
> **Response:**
>
> Yes, we will upload the training and inference code as well as the pre-trained models of MelGen and the ASR.

---

> > ### Comment · Reviewer_nskp · 2023-11-16
> >
> > Thank you for the response. Looking forward to the HiFi-GAN experiments and the code release.
> >
> > Regarding VoxCeleb2, other colleagues and I have experienced this. This almost certainly means that Google has blocked your IP and throttled your download speed. I would suggest downloading using a different IP via a different router or a VPN, this usually works. KB/s speeds are not normal from YouTube. In any case, I understand that the timeline is tight and will not lower the score if you can't provide VoxCeleb2 results.

---

> > > ### Author Response · Authors · 2023-11-19
> > >
> > > 1. *HiFi-GAN*
> > >
> > > We compare here the metrics for LRS3 as computed on speech signals generated using HiFi-GAN and DiffWave as the vocoders. Both vocoders used the exact same mel-spectrograms generated by LipVoicer.
> > >
> > > | vocoder  | WER   | STOI-Net | DNS-MOS | LSE-C | LSE-D |
> > > |----------|-------|----------|---------|-------|-------|
> > > | DiffWave | 21.4% | 0.92     | 3.11    | 6.239 | 8.266 |
> > > | HiFi-GAN | 21.9% | 0.93     | 3.19    | 6.308 | 8.166 |
> > >
> > > We see in this table that both options that use two different vocoders yield state-of-the-are results.
> > > This clearly indicates that the main and critical source of improvement is the mel-spectrograms generated by our method.
> > >
> > > 2. *VoxCeleb2*
> > >
> > > We thank the reviewer for the advice. We are downloading the videos on a remote server where the GPUs run which is administered and moderated by our institution. Consequently, we cannot unfortunately use workarounds such as a VPN.
> > > However, we will very soon post our results for training LipVoicer on LRS2+LRS3.

---

> > > > ### Comment · Reviewer_nskp · 2023-11-20
> > > >
> > > > Thank you for the HiFi-GAN ablation. Looking forward to the LRS2+LRS3 results.

---

> ### Author Response · Authors · 2023-11-21
>
> Here are the results on the test set of LRS3, when the training set is LRS2+LRS3
>
> | Training data | WER$\downarrow$   | STOI-Net$\uparrow$ | DNS-MOS$\uparrow$ | LSE-C$\uparrow$ | LSE-D$\downarrow$ |
> |---------------|-------|----------|---------|-------|-------|
> | LRS3          | 21.4% | 0.92     | 3.11    | 6.239 | 8.266 |
> | LRS2+LRS3     | 21.2% | 0.93     | 3.15    | 6.308 | 8.214 |
>
> This tables shows that the model's performance scales with the amount of training data.

---

> > ### Comment · Reviewer_nskp · 2023-11-21
> >
> > Thank you for these new results. Would be great to include these in the paper of course. While the performance improvement is not terribly impressive I understand this experiment was run in a rush due to the tight rebuttal time so it's understandable. Maybe with some hyperparameter tuning it can get better. Also, would still be great to see VoxCeleb2 results in the camera-ready version. In any case, given that you have addressed all of my concerns, your response to other reviewers seems adequate, and the paper was already strong in my opinion, I'm happy to raise my score.

---

> > > ### Author Response · Authors · 2023-11-21
> > >
> > > We thank the reviewer for the positive feedback, and as agreed all of the new results including the VoxCeleb2 experiment will be added to the camera-ready version

---

### Official Review · Reviewer_gU1H · 2023-10-29

**Soundness:** 2 fair
**Presentation:** 3 good
**Contribution:** 2 fair
**Rating:** 5
**Confidence:** 4

**Summary:**

This paper proposes a method to generate a natural-sounding speech from silent video called LipVoicer. The method is different from previous work in a two key ways: (1) the proposed method uses a lip reading model during inference to generate guidance for the generation model, (2) the generative model is based on a diffusion model. The model is trained on LRS2 and LRS3 datasets, which contains challenging examples from near in-the-wild conditions. The proposed system significantly outperforms the baselines.

The two key ideas actually appeared in accepted recent/concurrent papers, and authors acknowledge these works. Lip reading-based text guidance is proposed in (Kim et al., ICASSP 2023), although it is not exactly the same in that this paper uses the text guidance during inference, whereas Kim et al. uses the guidance during training. The use of diffusion-based model for the lip-to-speech task has been proposed in (Choi et al, ICCV 2023a). Authors are not required to compare their own work to that paper under ICLR rules.

**Strengths:**

- The key ideas are reasonable, and well-engineered combination of proven methods.
- The use of pre-trained state-of-the-art lip reading model significantly lowers the WER significantly compared to existing methods.
- The diffusion model generates natural-sounding output, according to the qualitative results reported.

**Weaknesses:**

- It is not clear if the performance improvement comes from the key improvements, or the replacement of the vocoder, which can be seen as a post-processing step rather than a key part of the algorithm. It is well known that DiffWave produces much more natural-sounding output compared to the Griffin-Lim algorithm used by the previous works.
- The authors request subjective assessors to rate Intelligibility, Naturalness, Quality and Synchronisation, but it is not clear what the difference between Naturalness and Quality are. There is a screenshot of the evaluation page in the appendix, but it does not make it clear what 'quality' means.
- The baseline models appear to be using pre-trained model weights. However, the models are not trained on the same data, so the results cannot be compared directly.
- The method appears to apply Guided-TTS techniques to the problem of lip-to-speech. Although this is well engineered, in my opinion this work is better suited to speech or CV conference compared to ICLR.

**Questions:**

- It is not clear why the addition of text guidance helps sync performance.
- If lip reading networks are used, what is the advantage of the proposed system over a cascaded lip reading + TTS system apart from obtaining duration prediction from sync.

---

> ### Author Response · Authors · 2023-11-16
>
> 1. *It is not clear if the performance improvement comes from the key improvements, or the replacement of the vocoder, which can be seen as a post-processing step rather than a key part of the algorithm. It is well known that DiffWave produces much more natural-sounding output compared to the Griffin-Lim algorithm used by the previous works.*
>
> **Response:**
>
> As Figures 2 & 3 in the Appendix show, LipVoicer recovers mel-spectrograms that look like the mel-spectrogram of natural speech. The competitors, however, fail to do so and are missing crucial pitch information and hence sound unintelligible and metallic. Even an excellent vocoder cannot compensate for degraded input mel-spectrograms. Moreover, we trained the vocoder on mel-spectrograms which were extracted from natural speech signals, and not on the mel-spectrograms that we generated with LipVoicer. This gives a strong qualitative indication that the source of the improvement comes mainly from the quality of the mel-spectrogram reconstructed by LipVoicer.
> We note that we compare favourably to SVTS that uses a modern neural vocoder, specifically Parallel WaveGan, and not Griffin-Lim. In general, Griffin-Lim is rarely used nowadays in recent papers in the text- and lip- to speech techniques, since more advanced vocoders currently exist.
>
> 2. *The authors request subjective assessors to rate Intelligibility, Naturalness, Quality and Synchronisation, but it is not clear what the difference between Naturalness and Quality are. There is a screenshot of the evaluation page in the appendix, but it does not make it clear what 'quality' means.*
>
> **Response:**
>
> Quality is the overall quality of the audio. It is equivalent to the single value of MOS that most papers report. The other metrics try to break down the quality into different components to understand the specific pros and cons of each model. Naturalness refers to how natural the audio is, for example in terms of the voice and prosody. A speech signal can achieve high Naturalness, but at the same time low Quality score since, for instance, some words are not clearly pronounced or they are not well synchronised with the video.
>
> 3. *The baseline models appear to be using pre-trained model weights. However, the models are not trained on the same data, so the results cannot be compared directly.*
>
> **Response:**
>
> We trained the baselines on LRS3 and LRS2, and we thank the reviewer for drawing our attention that we forgot to mention it in the text. Our comparisons are not based on the pre-trained models available on the project pages of the baselines. We will clarify this in the text.
>
> 4. *The method appears to apply Guided-TTS techniques to the problem of lip-to-speech. Although this is well engineered, in my opinion this work is better suited to speech or CV conference compared to ICLR.*
>
> **Response:**
>
> We would like to draw the reviewer’s attention that papers of similar or related flavour have been published at ML conferences. For example, [1-3] were published at ICLR, [4] was published at NeurIPS and [5] was published at ICML.
>
> [1] Felix Kreuk, Gabriel Synnaeve, Adam Polyak, Uriel Singer, Alexandre Défossez, Jade Copet, Devi Parikh, Yaniv Taigman, and Yossi Adi. AudioGen: textually guided audio generation. The International Conference on Learning Representations (ICLR). 2023.
>
> [2] Haoyue Cheng, Zhaoyang Liu, Wayne Wu and Limin Wang. Filter-recovery network for multi-speaker audio-visual speech separation. The International Conference on Learning Representations (ICLR). 2023.
>
> [3] Kai Li, Runxuan Yang and Xiaolin Hu. An efficient encoder-decoder architecture with top-down attention for speech separation. The International Conference on Learning Representations (ICLR). 2023.
>
> [4] Minsu Kim, Joanna Hong, and Yong Man Ro. Lip to speech synthesis with visual context attentional gan. Advances in Neural Information Processing Systems, 34:2758–2770, 2021.
>
> [5] Eliya Nachmani, Yossi Adi, and Lior Wolf. Voice separation with an unknown number of multiple speakers. Proceedings of the 37th International Conference on Machine Learning, 2020.

---

> ### Author Response · Authors · 2023-11-19
>
> We compare here the metrics for LRS3 as computed on speech signals generated using HiFi-GAN and DiffWave as the vocoders. Both vocoders used the exact same mel-spectrograms generated by LipVoicer. HiFi-GAN is widely used in the literature, including in (Choi et al, ICCV 2023a).
>
> | vocoder  | WER   | STOI-Net | DNS-MOS | LSE-C | LSE-D |
> |----------|-------|----------|---------|-------|-------|
> | DiffWave | 21.4% | 0.92     | 3.11    | 6.239 | 8.266 |
> | HiFi-GAN | 21.9% | 0.93     | 3.19    | 6.308 | 8.166 |
>
> We see in this table that both options that use two different vocoders yield state-of-the-are results.
> This clearly indicates that the main and critical source of improvement is the mel-spectrograms generated by our method.

---

### Official Review · Reviewer_x1vF · 2023-10-31

**Soundness:** 4 excellent
**Presentation:** 4 excellent
**Contribution:** 2 fair
**Rating:** 6
**Confidence:** 4

**Summary:**

This paper proposes LipVoicer, which incorporates the text modality by predicting the spoken text using a pre-trained lip-reading network and conditioning a diffusion model on both the video and the extracted text. To utilize the text modality into the diffusion model, the authors apply classifier-guidance mechanism, where a pre-trained automatic speech recognition (ASR ) serves as the classifier. The results demonstrate the effectiveness of LipVoicer in producing natural, synchronized, and intelligible speech.

**Strengths:**

1. LipVoicer greatly improves the intelligibility of the generated speech and outperforms existing lip-to-speech baselines on challenging datasets, demonstrating its superior performance.

2. The paper provides detailed implementation details, making it easier for others to reproduce and further improve upon the LipVoicer method.

3. By introducing a pre-trained ASR model, this paper realizes a good application of classifier-guidance diffusion model in lip2speech task.

**Weaknesses:**

1. After listening to Demo page, it is found that the gap between different models is mainly in sound quality. The baselines are too weak in sound quality. However, the problem of sound quality can be solved by many existing generative models based on VAE/GAN/FLOW model. If the sound quality problem of baselines is solved, the advantage of the model proposed in this paper may not be so great.

2. In previous studies, a very important motivation for lip2speech tasks was to dispense with text modality (otherwise, this task can be transformed into lipreading+TTS), because 80% of the world's languages have no written text. However, this paper still depends on the text modality, so it is difficult to give a high score to this article.

**Questions:**

None

---

> ### Author Response · Authors · 2023-11-16
>
> 1. *After listening to Demo page, it is found that the gap between different models is mainly in sound quality. The baselines are too weak in sound quality. However, the problem of sound quality can be solved by many existing generative models based on VAE/GAN/FLOW model. If the sound quality problem of baselines is solved, the advantage of the model proposed in this paper may not be so great.*
>
> **Response:**
>
> One of the main contributions of our paper **exactly is** the improvement in speech quality. The lip-to-speech task is more than just generating high quality audio. We need to produce a speech signal that follows the words spoken in a silent video with unrestricted vocabulary, while achieving naturalness, appealing timing, intonation and vocal features that reflect the appearance of the speaker. This very information must be extracted from the silent video. As Figures 2 & 3 in the Appendix show, LipVoicer recovers mel-spectrograms that look like the mel-spectrogram of natural speech and follow the words likely to be spoken in the video. The competitors, however, fail to do so and are missing crucial pitch information and hence sound unintelligible and metallic.
>
> 2. *In previous studies, a very important motivation for lip2speech tasks was to dispense with text modality (otherwise, this task can be transformed into lipreading+TTS), because 80% of the world's languages have no written text. However, this paper still depends on the text modality, so it is difficult to give a high score to this article.*
>
> **Response:**
>
> Our problem is fundamentally different from a lip-reading+TTS setup. The generated speech must be synchronised with the video, match the speaker appearance and convey prosody. Whereas lip-reading+TTS cannot achieve these objectives, we show how to successfully incorporate the text in a way which does manage to satisfy all of these tasks. Our claim is that the text is important, and should not be dispensed so fast. Importantly, we utilise it without resorting to transcribed videos by humans. The way that we acquire (and exploit) the textual modality is exactly one of the virtues of  LipVoicer. We suggest using a lip-reader, and therefore the text comes for free. We show in our experiments and the provided speech samples that text plays a major part in the success of our method.
>
> Regarding the fact that 80% of the world's languages have no written text, while this might be true, the vast majority of speakers do speak languages with written text. While our solution does not solve all languages, it is relevant to the vast majority of the population and as such we believe this is significant enough to be of value to the community.

---

### Official Review · Reviewer_eBPV · 2023-11-04

**Soundness:** 3 good
**Presentation:** 3 good
**Contribution:** 3 good
**Rating:** 5
**Confidence:** 3

**Summary:**

This work proposes to perform the lip-to-speech task by incorporating the predicted text to guide the diffusion model based learning process. Experiments on the large scale LRS2 and LRS3 show its superiority over others. The results are indeed appealing.

**Strengths:**

The general structure is clear. The method is simple in general. It’s easy to follow. The performance is good, with a large margin over other methods. It’s also a nice try to include the predicted text into the learning process.

**Weaknesses:**

(1) I am a little confused with fig1.a. The output of the lipreading module is the predicted text. The output of the ASR modules is also the predicted text. There should be no connections from the output of the predicted text to the ASR module? The ASR module should take the output of MelGen as input? without the text predicted from LR module?
(2) Lip2speech (Kim et al.(2023)) takes the ground-truth text as input to constrain the learning process and has shown the success of the role of text modality in this task. In this paper, the work uses the predicted text instead of the ground truth as Lip2Speech. But the manner is similar to Kim et al.(2022). So, besides using the predicted text with an existing method, is there some new contributions in the view of methodology?

**Questions:**

(1) I am a little confused about the fig.1(a) as described above.
(2) The modules and manners in the framework seems to be not new in the view of methodology, with the lipreading module, MelGen, text alignment manners already proposed by other works. Could the authors give a clarification of the contributions? Maybe I miss something?
(3) The performance using the predicted text is already very appealing, but the involved lip reading model are almost the best two ones at present, with WER=19% and 26%. if the lip reading performance has been a much low value, e.g. WER=50%, what would be the performance here like?

---

> ### Author Response · Authors · 2023-11-16
>
> 1. *I am a little confused with fig1.a.*
>
> **Response:**
>
> The inference procedure of LipVoicer involves the estimation of two noise terms. The first one, $\epsilon_{mg}$ is computed via classifier-free guidance using the MelGen module, whose inputs are the noisy mel-spectrogram $\mathbf{x}\_{t+1}$, the lip motion video and a face image. The second noise term is computed via classifier guidance. The ASR takes $\mathbf{x}\_{t+1}$ as an input, and computes the loss at the output with respect to the text predicted by the lip-reading network.
> We have updated the figure so it better reflects the methodology of our method. We thank the reviewer for this comment which has helped us improve the clarity of the Fig. 1.a. We have also added a detailed Algorithm in the Appendix for further clarification.
>
> 2. *In this paper, the work uses the predicted text instead of the ground truth as Lip2Speech. But the manner is similar to Kim et al.(2022). So, besides using the predicted text with an existing method, is there some new contributions in the view of methodology?*
>
> **Response:**
>
> We would like to draw the reviewer’s attention that LipVoicer utilises the **unsupervised text** at **inference time**, whereas Lip2Speech uses the **ground-truth text** for **training** in the form of an additional loss term. Integrating the textual modality with the audio at inference time is not straightforward, and it considerably changes the algorithmic approach that needs to be taken. The words in the textual modality bear no information about their timing, duration and intonation. As Tables 1-4 demonstrate,  by using the classifier guidance, LipVoicer manages to deploy the text fairly better than Lip2Speech despite using noisy text from the lip-reader. Moreover, the ablation study in Table 6 shows that when the text is not used (w2=0), the WER score drops drastically. However when w2 is adequately chosen, we manage to achieve both excellent WER **and** audio quality. Another important difference between LipVoicer and Lip2Speech is that in LipVoicer we manage to extract vocal features of the speaker and make the generated audio reflect the identity of the speaker, whereas Lip2Speech lacks this ability.
>
> 3. *The modules and manners in the framework seems to be not new in the view of methodology, with the lipreading module, MelGen, text alignment manners already proposed by other works*
>
> **Response:**
> Lipvoicer does indeed use existing algorithms as key components, but the key contribution and innovation is in the use of predicted text from lipreading in the speech generation process. We show that we can use a lip-reading network to acquire the textual modality without any additional supervision, and use the extracted text in the speech generation process in a way which significantly increases the intelligibility while maintaining speech quality.
> A benefit of the design of LipVoicer is that it is highly modular - every single block in the architecture, e.g. MelGen/lip-reader/ASR can be substituted by a more advanced module. As our ablation studies show, this leads to improvement of the generated speech.
> Additionally, unlike previous methods that align text, we use an ASR and not a phoneme classifier, which is cumbersome and harder to train. For example, LRS3 comprises several dialects of the English language. Training a phoneme classifier on such a dataset requires a comprehensive dictionary that translates all the words to phoneme sequences according to each dialect, and the probability of occurrence for each word-phonemes pair. Subsequently, a model that aligns the phonemes to the spectrogram frames must be trained. Compared to this, an ASR is much easier to use and does not require in-depth knowledge of phonemes or the dialects present in the dataset.
>
> 4. *The performance using the predicted text is already very appealing, but the involved lip reading model are almost the best two ones at present, with WER=19% and 26%. if the lip reading performance has been a much low value, e.g. WER=50%, what would be the performance here like?*
>
> **Response:**
>
> Our main claim is that the text modality, even if it is not perfect, can be of great help to audio generation from video. Therefore we do not argue that our method will perform well with a bad lip-reader, we rather say that the existing lip-readers can greatly boost performance. This is further corroborated in the ablation study we show in Table 7. A lip-reader with WER=32.3% is already not so good, and we show that LipVoicer is adversely affected by it. The better the lip-reader gets, the better our model performs.

---

### Author Response · Authors · 2023-11-16

We wish to thank the reviewers for their time and effort, as well as their constructive and insightful feedback. Additionally, we are glad that  Reviewer nskp has found our method to yield state-of-the art results, and that the generated speech is of high quality and sounds natural and intelligible.

Following Reviewer eBPV’s comment, we have updated Fig. 1.a to improve the illustration of the inference process. We also introduce Algorithm 1 in the Appendix that summarises the steps of the inference.
Addressing a question that was asked by two reviewers, we want to clarify that the lip-to-speech task is not equivalent to lip-reading+TTS even in the presence of text. The generated speech must be synchronised with the video, match the speaker appearance and convey prosody. While lip-reading+TTS cannot achieve these objectives, we show how to successfully incorporate the text in a way which does manage to satisfy all of these tasks.

Lastly, another issue that was raised by reviewers gU1H and nskp was the choice of the vocoder. We chose to use DiffWave as our vocoder since we had already used its architecture for MelGen, so applying a DiffWave vocoder was immediate. As Figures 2 & 3 in the Appendix show, LipVoicer recovers mel-spectrograms that look like the mel-spectrogram of natural speech. The competitors, however, fail to do so and are missing crucial pitch information and hence sound unintelligible and metallic. Even an excellent vocoder cannot compensate for degraded input mel-spectrograms. Moreover, we trained the vocoder on mel-spectrograms which were extracted from natural speech signals, and not on the mel-spectrograms that we generated with LipVoicer. This gives a strong qualitative indication that the source of the improvement comes mainly from the quality of the mel-spectrogram reconstructed by LipVoicer. We are now training a HiFi-GAN vocoder, which is used by many TTS and lip-to-speech frameworks. Once training is completed we will use it to synthesise test audio samples and upload the calculated metrics to this page.

---

> ### Author Response · Authors · 2023-11-20
>
> Dear reviewers, we hope we have addressed your questions, and we will be happy to discuss any questions that might follow

---

### Meta-Review · Area_Chair_CGM6 · 2023-12-09

**Metareview:**

This paper presents a method called LipVoicer to synthesise speech from silent videos, which incorporates the text modality by predicting the text using a pre-trained lip-reading model and conditioning a diffusion model on both the video and the extracted text. Experimental results demonstrate the effectiveness of the proposed method in producing natural, synchronized, and intelligible speech. While the performance is good, the novelty in methodology is moderate. There is also a discussion about whether the baseline is strong enough.

**Justification For Why Not Higher Score:**

The paper is borderline paper. See meta review.

**Justification For Why Not Lower Score:**

The paper could get lower score. See meta review.

---

### Decision · Program_Chairs · 2024-01-16

Accept (poster)